# The Effects of Annatto Tocotrienol Supplementation on Cartilage and Subchondral Bone in an Animal Model of Osteoarthritis Induced by Monosodium Iodoacetate

**DOI:** 10.3390/ijerph16162897

**Published:** 2019-08-13

**Authors:** Kok-Yong Chin, Sok Kuan Wong, Fadhlullah Zuhair Japar Sidik, Juliana Abdul Hamid, Nurul Hafizah Abas, Elvy Suhana Mohd Ramli, Sabarul Afian Mokhtar, Sakthiswary Rajalingham, Soelaiman Ima Nirwana

**Affiliations:** 1Department of Pharmacology, Faculty of Medicine, Universiti Kebangsaan Malaysia, Cheras 56000, Malaysia; 2Department of Anatomy, Faculty of Medicine, Universiti Kebangsaan Malaysia, Cheras 56000, Malaysia; 3Department of Orthopedics and Traumatology, Faculty of Medicine, Universiti Kebangsaan Malaysia, Cheras 56000, Malaysia; 4Department of Medicine, Faculty of Medicine, Universiti Kebangsaan Malaysia, Cheras 56000, Malaysia

**Keywords:** arthritis, antioxidant, inflammation, joint, vitamin E

## Abstract

Osteoarthritis is a degenerative joint disease which primarily affects the articular cartilage and subchondral bones. Since there is an underlying localized inflammatory component in the pathogenesis of osteoarthritis, compounds like tocotrienol with anti-inflammatory properties may be able to retard its progression. This study aimed to determine the effects of oral tocotrienol supplementation on the articular cartilage and subchondral bone in a rat model of osteoarthritis induced by monosodium iodoacetate (MIA). Thirty male Sprague-Dawley rats (three-month-old) were randomized into five groups. Four groups were induced with osteoarthritis (single injection of MIA at week 0) and another served as the sham group. Three of the four groups with osteoarthritis were supplemented with annatto tocotrienol at 50, 100 and 150 mg/kg/day orally for five weeks. At week 5, all rats were sacrificed, and their tibial-femoral joints were harvested for analysis. The results indicated that the groups which received annatto tocotrienol at 100 and 150 mg/kg/day had lower histological scores and cartilage remodeling markers. Annatto tocotrienol at 150 mg/kg/day significantly lowered the osteocalcin levels and osteoclast surface of subchondral bone. In conclusion, annatto tocotrienol may potentially retard the progression of osteoarthritis. Future studies to confirm its mechanism of joint protection should be performed.

## 1. Introduction

Osteoarthritis, a degenerative joint condition, is one of the most prevalent debilitating diseases affecting the elderly worldwide. It was estimated that 3.8% and 0.85% of the population worldwide suffered from knee and hip osteoarthritis respectively in 2010, constituting the 11th and the 38th largest global morbidity measured by disability-adjusted-life-years (DALYs) [1]. Overall, the DALYs of osteoarthritis were higher than colon/rectum cancer (44th), breast cancer (47th) and Alzheimer disease (49th) [2].

The development of osteoarthritis is hypothetically initiated by excessive mechanical loading on the weight-bearing joints, leading to the breakdown of the articular cartilage. The wear-and-tear products of the cartilage will trigger infiltration of the inflammatory cells into synovium, and the subsequent release of proinflammatory cytokines, such as tumor necrosis-alpha and interleukin-1 beta and interleukin-6. These cytokines will invoke inflammation of the joint tissues and the release of metalloproteinases and aggrecanases, causing further cartilage breakdown. When the self-perpetuating inflammation and cartilage wearing process overwhelms the reparative capacity of the chondrocytes, cartilage thinning and osteoarthritis tend to occur [3,4,5].

Another aspect of osteoarthritis often overlooked is the alternation in the subchondral bone compartment. Increased osteoclastic activity and bone resorption are often observed in the subchondral bone during the early stages of osteoarthritis. This is followed by increased bone and osteophyte formation, probably due to excessive mechanical loading on the subchondral bone [6,7]. Previous studies suggested that preventing the increased remodeling of the subchondral bone could slow down the progression of osteoarthritis [8,9].

The current therapeutic approach of osteoarthritis focuses on eliminating the pain suffered by the patients through paracetamol and non-steroidal anti-inflammatory agents [10]. The beneficial effects of joint protecting supplements like glucosamine, avocado/soybean unsaponifiables remain debatable [11]. As local chronic inflammation plays an integral part in the pathogenesis of osteoarthritis, natural supplements with anti-inflammatory properties could prevent the development of joint degeneration. Tocotrienol, a subfamily of vitamin E, has demonstrated promising anti-inflammatory properties [12,13]. Tocotrienol can be divided into four homologues (alpha, beta, gamma, delta) and they usually exist together with alpha-tocopherol, the most prevalent vitamin isoform in natural [14,15]. As our body prioritizes the absorption and retainment of alpha-tocopherol, its presence has been shown to reduce the absorption, bioavailability and biological effects of tocotrienol [16,17,18]. Tocotrienol mixture derived from annatto beans contains approximately 90% delta-tocotrienol, 10% gamma-tocotrienol and without any detectable amount of alpha-tocopherol [19]. This prompted us to study the effects of tocotrienol on the joint without the interference of alpha-tocopherol.

Therefore, this study aimed to investigate the effects of annatto tocotrienol on the joints of a rat model of osteoarthritis induced by monosodium iodoacetate. The secondary objective was to determine the effects of annatto tocotrienol on changes in the subchondral bone.

## 2. Materials and Methods

### 2.1. Materials

Annatto tocotrienol (lot number: 17CA3-A3-70), consisting of 84% δ-tocotrienol and 16% γ-tocotrienol, was donated by American River Nutrition (Hadley, MA, USA) for the purpose of this study. It was diluted with olive oil (Bartolini Emilio, Arrone Terni, Italy) in the ratio of 1:10 prior to administration. The diluted annatto tocotrienol was prepared weekly and stored at 4 °C in capped dark bottles before use. Monosodium iodoacetate (MIA) was purchased from Sigma-Aldrich (St. Louis, MO, USA) and dissolved in normal saline (4 mg in 50 µL) prior to administration.

### 2.2. Animals and Treatment

Three-month-old male Sprague-Dawley rats (n = 30) were purchased from the Laboratory Animal Resource Unit, Universiti Kebangsaan Malaysia (Kuala Lumpur, Malaysia). They were kept at Animal Laboratory of Anatomy Department, Universiti Kebangsaan Malaysia (Cheras, Malaysia) under standard condition (27 °C, 12:12 light cycle) and given standard rat chow and tap water ad libitum throughout the experiment. After a week of acclimatization, they were randomly assigned to five groups (n = 6/group). All groups except the sham were injected with MIA intraarticularly once at the right knee of the hindlimbs at the beginning of the study (day 0). The sham group received an equal volume of normal saline injection at the right knee on the same day. Three groups receiving MIA injection were administrated annatto tocotrienol orally at the dose of 50, 100, 150 mg/kg/day a day after the MIA injection. The doses were based on previous studies, whereby annatto tocotrienol demonstrated skeletal effects at doses between 60–100 mg/kg [20]. The sham and MIA group were supplemented with an equal volume of olive oil. The rats were treated six days a week for five weeks, after which they were sacrificed under anesthesia. Their blood samples were collected from the tail vein at the beginning of the study and through cardiac puncture upon sacrifice. The knee joints were harvested for histological analysis. The protocol of this study had been reviewed and approved by Universiti Kebangsaan Malaysia Animal Ethics Committee (code: FF-2016-119).

### 2.3. Histological Analysis

Soft tissue was removed from the tibial-femoral joints and the samples were immersed in 10% formalin with ethylenediaminetetraacetic acid (5.5% w/v) for two months for decalcification. Then, the joint tissues were processed into paraffin blocks and sectioned longitudinally at the thickness of 5 µm using a microtome (Leica Biosystem, Leica RM2235, Nussloch, Germany). The slides were rinsed with xylene to remove the paraffin, rehydrated, stained with 0.1% fast green solution (5 min), rinsed with 1% acetic acid (10 s), stained with 0.1% safranin O solution (20 min), dehydrated with alcohol, and mounted. The slides were then scored by a blinded anatomist. The aspects of the joint evaluated were pannus formation, synovial hyperplasia, infiltration of inflammatory cells and cartilage erosion. The scoring system for each aspect was 0, normal; 1, mild changes; 2, moderate changes; 3, severe changes [21].

The rehydrated slides were also stained with hematoxylin (10 min), rinsed with 1% acetic acid (10 s), stained with eosin (5 min) and dehydrated with alcohol before being mounted for subchondral bone observation. Subchondral bone below the articular cartilage of the tibia was evaluated using a quantitative stereological method with the aid of a Weibel grid consisting of 21 lines and 42 points [22]. The number of lines intersecting with trabecular bone was calculated as bone surface. The number of lines intersecting with the bone cells/osteoid was calculated as the bone cell/osteoid surface. The number of points intersecting with trabecular bones was calculated as the bone volume. The number of points intersecting with osteoid was calculated as the osteoid volume [23]. The parameters measured include osteoblast surface (Ob.S/BS; %), osteoclast surface (Oc.S/BS; %), eroded surface (ES/BS; %), osteoid surface (OS/BS; %) and osteoid volume (OB/BV; %).

All the slides were observed using a light microscope (Nikon Eclipse 80i, Tokyo, Japan) equipped with an imaging software (Media Cybernetics Image Pro-Plus, Rockville, MD, USA). At least four slides were evaluated for each rat.

### 2.4. Biochemical Evaluation

The blood collected using plain tube was centrifuged (3000 rpm for 10 min at 4 °C) to isolate the serum, which was stored at −70 °C until analysis. Two serum cartilage remodeling markers (cartilage oligomeric matrix protein/COMP (cartilage oligomeric matrix protein) and hyaluronic acid) and two bone remodeling markers (formation: osteocalcin; resorption: C-terminal telopeptide of type 1 collagen/CTX-1) were evaluated using enzyme-linked immunoassay kits (Finetest, Wuhan, China) following manufacturer’s instructions.

### 2.5. Statistical Analysis

Normality of the data was analyzed using Shapiro-Wilk test. All data, except histological score of the joint, were normally distributed. The histological score of the joint was evaluated using the Kruskal-Wallis test and Mann-Whitney U-test with Bonferroni adjustment (*p*-value <0.005 as significant) was used for pairwise comparison. Bone cellular histomorphometrical results were analyzed using one-way analysis of variance (ANOVA) with Tukey post hoc pairwise comparison. The body weight and biochemical parameters were analyzed using mixed-design ANOVA (time × group) with simple effect analysis. A *p*-value <0.05 was considered statistically significant unless stated otherwise. The statistical analysis was performed using SPSS version 23.0 (IBM, Armonk, NY, USA).

## 3. Results

All experimental groups experienced a significant increase in body weight throughout the study period (*p* < 0.05) but there was no significant inter-group difference at each time point (*p* = 0.657) (Figure 1).

The micrographs showed that knee joints of the rats in sham group were well-stained and the cartilage layer was intact. The knee joints of the rats receiving MIA was denuded while the annatto tocotrienol treatment preserved the cartilage layer (Figure 2). Monosodium iodoacetate caused degeneration of the articular cartilage of the rats, based on higher scores of pannus formation, synovial hyperplasia, infiltration of inflammatory cells and erosions compared to the sham group (*p* < 0.001). Treatment with annatto tocotrienol at 50 mg/kg/day did not improve the degenerative changes significantly. At 100 mg/kg/day, annatto tocotrienol resulted in less joint degeneration, as indicated by significantly lower scores across all histological aspects (*p* < 0.001). However, at 150 mg/kg/day, annatto tocotrienol was associated with significantly lower scores for synovial hyperplasia and erosions only (*p* < 0.001) (Figure 3, Appendix A).

For serum biomarkers of cartilage remodeling, COMP showed a significant time × group effects (*p* < 0.001). Simple effects analysis revealed that COMP level was lower in all annatto tocotrienol-treated groups compared to MIA group. The COMP level was also lower in rats treated with annatto tocotrienol at the dose of 100 and 150 mg/kg/day compared to 50 mg/kg/day. Serum hyaluronic acid showed an increase over the five-week period (*p* < 0.001 for time). A significant time × group effects (*p* = 0.040) was also observed, whereby the serum level of hyaluronic acid was significantly higher in the MIA group compared to the annatto tocotrienol groups. There was no significant difference in serum hyaluronic acid level across the annatto tocotrienol-treated groups (*p* > 0.05) (Figure 4A,B, Appendix A).

The cellular histomorphometrical indices for subchondral bone generally did not show significant changes with time and between groups (*p* > 0.05). Only rats treated with annatto tocotrienol at 150 mg/kg/day showed a significant reduction in Oc.S/BS post-treatment compared to MIA group (*p* < 0.05). All rats had higher ES/BS compared to the sham group (*p* < 0.05) (Figure 5, Appendix A).

For serum osteocalcin (bone formation marker), there was a significant time × group effect (*p* < 0.001). The MIA group showed a significantly lower serum osteocalcin level compared to the sham group (*p* < 0.05). Annatto tocotrienol at 50 mg/kg/day was shown to increase the osteocalcin level significantly compared to the MIA group (*p* < 0.05). However, at higher doses of annatto tocotrienol (100 and 150 mg/kg/day), the osteocalcin levels were significantly lower compared to rats treated with 50 mg/kg/day. Rats treated with 150 mg/kg/day had the lowest osteocalcin level among all the groups (*p* < 0.05). For serum CTX-1 levels, the time × group effect was not significant (*p* = 0.191) (Figure 4C,D, Appendix A).

## 4. Discussion

In this study, we examined the effects of annatto tocotrienol on the cartilage and subchondral bone of the knee joints in murine models. Osteoarthritis in the rats was induced by a known glycolysis inhibitor, MIA, which causes apoptosis of the chondrocytes and degradation of the cartilage [24]. The subsequent release of cartilage breakdown products can trigger localized inflammation of the joint. The self-perpetuating inflammation will cause more cartilage to breakdown [3,4,5]. These changes were observed in the MIA-treated rats. The increased scores for synovial hyperplasia and infiltration of inflammatory cells in the MIA-treated rats indicated inflammation of the synovium. Increased cartilage erosion in the osteoarthritic rats was the result of cartilage breakdown. The formation of fibrous pannus over the cartilage was hypothesized to be a failed reparative mechanism during osteoarthritis [25]. The products of cartilage breakdown, such as COMP and hyaluronic acid, were also found to be higher in MIA-treated rats compared to the sham group, comparable to observation in previous MIA osteoarthritis models [26,27].

Annatto tocotrienol, particularly at 100 mg/kg/day was able to retard the degenerative changes of the joint, as indicated by the lower scores for all histological aspects evaluated. The results of the current study were compared with studies using vitamin E/alpha-tocopherol [28]. Studies reported that the level of antioxidants, like vitamin E, was higher in the synovial fluid of patients with early-stage osteoarthritis but lower in late-stage osteoarthritis [29,30,31]. This could imply the mobilization of vitamin E in the body to overcome the oxidative stress in the joint, and its exhaustion as the disease progresses. Supplementation of alpha-tocopherol (source not indicated; 600 mg/kg, three times a week for 8 weeks) was shown to preserve the extracellular matrix and collagen fibrin sheet of the cartilage of rats administered with MIA [32]. Supplementation of alpha-tocopherol (DSM Nutritional Products, Fort Worth, TX, USA; 400 IU for 55 days) in dogs with osteoarthritis induced by cranial cruciate ligament transection prevented the formation of cartilage lesion at the femoral condyles [33]. In this study, annatto tocotrienol at 50 mg/kg/day was insufficient to prevent the degenerative changes at the joint. However, only the scores of synovial hyperplasia and cartilage erosion were significantly lowered by a higher dose of annatto tocotrienol (150 mg/kg/day). It should be noted that one of the rats fed with annatto tocotrienol 150 mg/kg/day showed moderate cartilage degeneration, thus it might have affected the overall histological scores of the group.

Osteoarthritis also affects the subchondral bone due to the abnormal mechanical loading. The abnormal mechanical loading will cause micro-fractures at the subchondral bone and the associated increased reparative remodeling activities. At the later stages, the bone remodeling activity will skew towards formation, leading abnormal bone formation and osteophytes [34,35]. Therefore, in this study, the cellular histomorphometric indices of the subchondral bone were examined. The ES/BS was increased in the MIA-treated rats compared to sham group. The corresponding increase in the Oc.S/BS was also observed despite a lack of statistical significance. These observations suggest the increase in bone resorption activity due to osteoarthritis. However, the level of bone remodeling markers, i.e., osteocalcin and CTX-1 did not change significantly with time in the osteoarthritis group receiving MIA. In fact, the CTX-1 level at the end of the treatment period was significantly lower in the MIA group compared to all other groups. A previous study among patients with osteoarthritis showed that a high bone remodeling (high procollagen 1 N-terminal and high CTX) was associated with less cartilage loss and vice versa. However, it is not certain that this finding could explain the observation of this study [36]. Other studies suggested that bone biomarkers like osteopontin and bone sialoprotein are more relevant in the pathogenesis of osteoarthritis [37]. Studies also found that CTX-II correlated better with disease progression of osteoarthritis [38,39].

Annatto tocotrienol did not alter the cellular histomorphometric indices of the subchondral bone significantly. Only the rats treated with annatto tocotrienol at 150 mg/kg/day showed a significant reduction in Oc.S/BS. Interestingly, annatto tocotrienol at 50 mg/kg/day triggered a significant elevation of osteocalcin level compared to the MIA group and rats treated with tocotrienol at higher doses. Tocotrienol mixture at 60 mg/kg/day was known to increase bone formation and mineral deposition in normal rats and various models of osteoporosis [40,41,42]. Since the increase in the Ob.S/BS was not associated with a parallel rise in the osteocalcin levels of the rats treated with 50 mg/kg/day annatto tocotrienol, it was suggested that annatto tocotrienol did not affect proliferation of osteoblasts but increased the bone formation activity of the existing osteoblasts. This was highlighted by a recent in vitro study, whereby annatto tocotrienol increased the differentiation and bone formation activity of MC3T3-E1 preosteoblasts without influencing its proliferation [43]. An increase in bone formation activity could possibly lead to the formation of osteophytes in the long run, thus this effect could be undesirable for patients with osteoarthritis. However, coupled with the high CTX-1 levels in the rats receiving 50 mg/kg/day annatto tocotrienol, we postulate that this is a sign of high bone turnover and not high bone formation per se. In contrast, annatto tocotrienol at 100 mg/kg/day did not increase the osteocalcin level, and at 150 mg/kg/day suppressed the osteocalcin level, although both doses did not suppress the increase in CTX-1 levels.

Overall, this study showed that annatto tocotrienol has joint protective effects in osteoarthritis. This was achieved by reducing cartilage degradation and subchondral bone remodeling. Previous studies have suggested that inflammation and oxidative stress were the key factors triggering the development of osteoarthritis [44]. Tocotrienol was shown to possess anti-inflammatory and antioxidant properties in the musculoskeletal system [45,46]. We compared our results with studies using vitamin E/alpha-tocopherol due to the paucity of data in this regard with tocotrienol. In experimental animals, the joint protective effects of alpha-tocopherol were observed in tandem with decreased inflammatory cytokines and oxidative stress markers [32,33]. In a recent clinical trial, Tantavisut et al. showed that vitamin E (400 IU) for two months reduced malondialdehyde and antioxidant capacity in the synovial fluid and blood of osteoarthritis patients [47]. At the same time, a reduction in synovial tissue stained cells with nitrotyrosine and inflammatory cells was observed in vitamin E-supplemented patients [47]. This corroborated with the previous findings of Bhattarcharya et al., whereby vitamin E (200 mg/day) for three months elevated the activity of serum antioxidant enzymes while decreasing the malondialdehyde level [48]. In future studies, it is worthwhile to compare the anti-inflammatory and antioxidant activities of tocotrienol with alpha-tocopherol in osteoarthritis.

Nevertheless, this study is not without its limitations. The concentration of tocotrienol was not tested in the synovial fluid, so its deposition in the knee joints was not known. The histological sections chosen were close to the shaft-axis of the bone, although the exact depth was not determined. Thus, it might cause biases in histological assessment. Immunostaining for apoptotic and catabolic markers of the joint was not performed, thus the exact mechanism of action of tocotrienol on the joint could not be ascertained. Similarly, identification cell types in the assessment of histological assessment was based solely on bone morphology without the aid of special staining like tartrate-resistant acid phosphatase (for osteoclasts) and CD markers (for inflammatory cells), so errors in estimation might occur. Serum hyaluronic acid is not a specific marker for osteoarthritis. However, a prospective study has established that serum hyaluronic acid correlated well with the progression of osteoarthritis defined radiographically across 5 years [49]. For subchondral bone evaluation, micro-computed tomography is the best instrument to measure the skeletal microarchitecture, but it is not available to the researchers at the time of the study. This study only tested the effects of annatto tocotrienol in male rats so its effects on female rats remains uncertain. Epidemiological data reveals that the prevalence of knee osteoarthritis is higher in women than men, especially after menopause [50]. Experimental studies showed that estrogen deficiency may exacerbate the progression of osteoarthritis [51]. Hence, the effects of annatto tocotrienol on bone and joint should be validated in a model of aged female rats mimicking the high-risk group in humans.

## 5. Conclusions

Annatto tocotrienol exerts joint protective effects by preventing cartilage degradation in an animal model of osteoarthritis. The optimal dose of annatto tocotrienol is suggested to be 100 mg/kg/day, as it showed the most favorable changes in the joint histology and serum cartilage markers. However, the effects of annatto tocotrienol on subchondral bone and its remodeling should be interpreted with caution, as it does not prevent the increase in serum bone resorption marker but lowers bone formation marker at high dose. Further studies should be performed to illustrate the mechanism of joint protection of annatto tocotrienol and its effects on functional parameters in osteoarthritis prior to application in humans.

## 6. Patents

The authors are currently applying for a patent based on the results of this work (file number: PI 2019001883).

## Figures and Tables

**Figure 1 ijerph-16-02897-f001:**
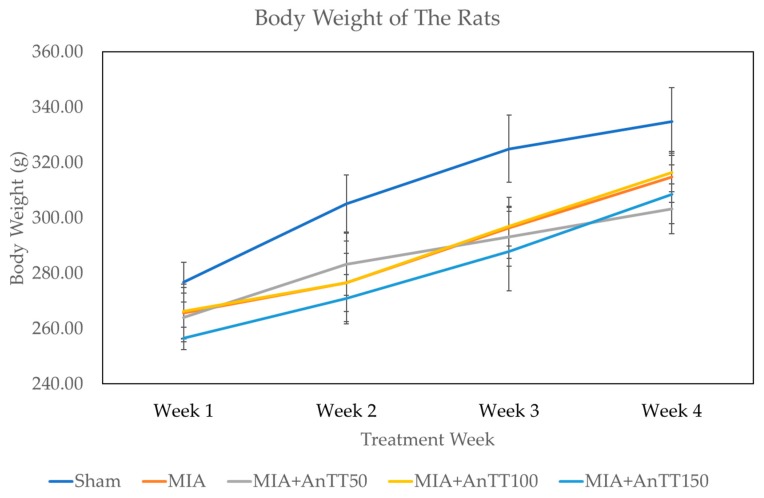
Body weight of the rats throughout the study period. The body weight was measured at the beginning of the week. No statistically significant intergroup difference at each time point was found.

**Figure 2 ijerph-16-02897-f002:**
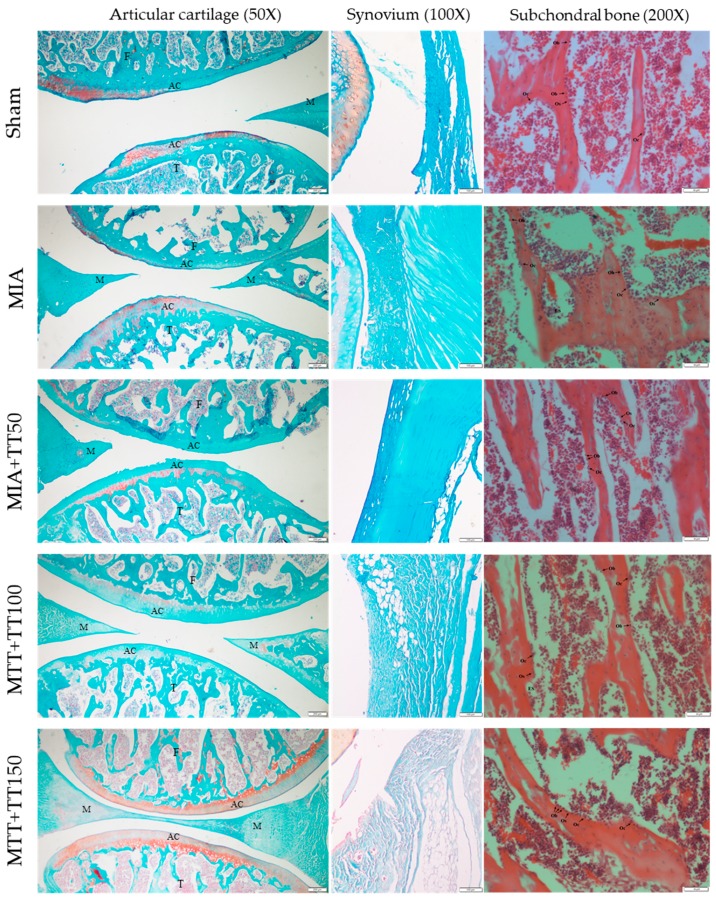
Micrographs of the knee joints of the rats receiving different treatments. Abbreviation: AC, articular cartilage; F, femur; M, meniscus; T, tibia; Ob, osteoblast; Oc, osteoclast; Os, osteoid; ES, eroded surface.

**Figure 3 ijerph-16-02897-f003:**
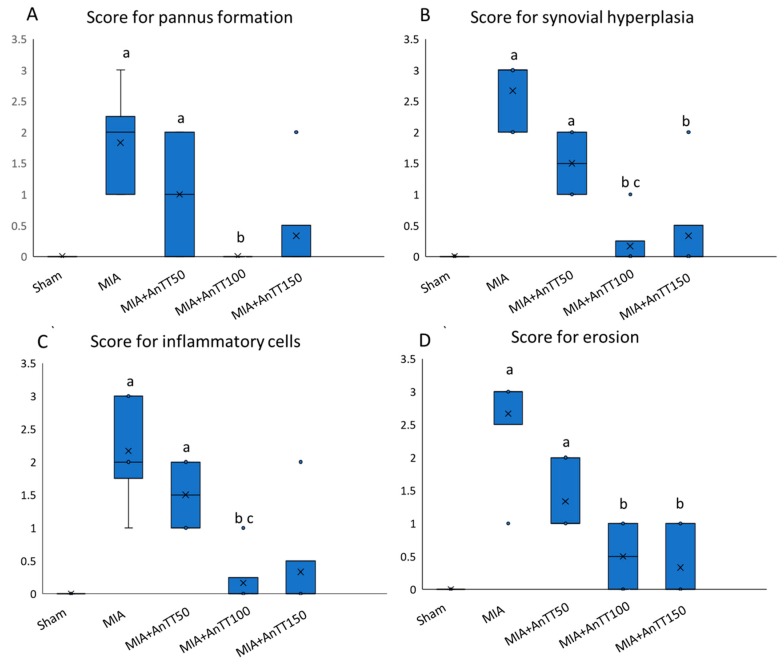
Joint histology score after treatment; the score for pannus formation (**A**); synovial hyperplasia (**B**); inflammatory cells (**C**); erosion (**D**). The data are shown as mean with standard error of the mean. The letter ‘a’ indicates a significant difference versus the sham group; ‘b’ versus the MIA group; ‘c’ versus the AnTT50 group. Abbreviation: AnTT, annatto tocotrienol. The symbol ‘x’ in the box plot represents the mean value, ‘∘’ represents the outliers, while the horizontal line in the box represents the median of the group.

**Figure 4 ijerph-16-02897-f004:**
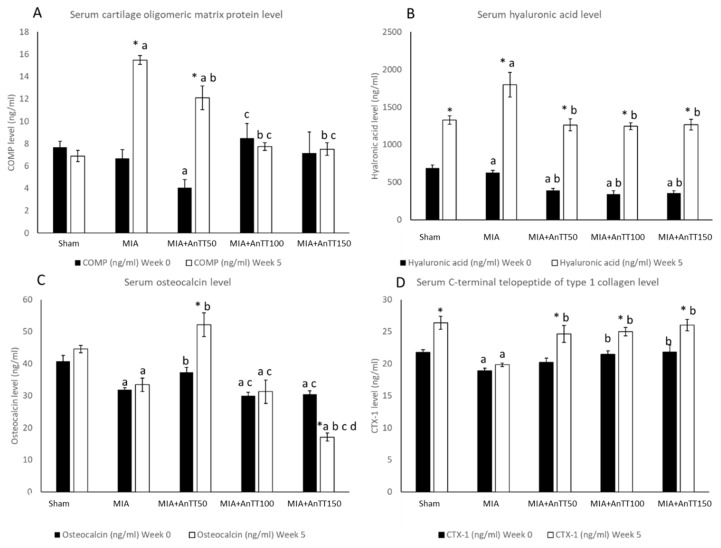
Joint and bone remodeling markers in the rats before and after treatment; serum cartilage oligomeric matrix protein level (**A**); serum hyaluronic acid (**B**); serum osteocalcin (**C**); Serum C-terminal telopeptide of type 1 collagen level (**D**). The data are shown as mean with standard error of the mean. The letter ‘a’ indicates significant difference versus the sham group; ‘b’ versus the MIA group; ‘c’ versus the AnTT 50 group; ‘d’ versus the AnTT 100 group; ‘*’ versus week 0. Abbreviation: AnTT, annatto tocotrienol.

**Figure 5 ijerph-16-02897-f005:**
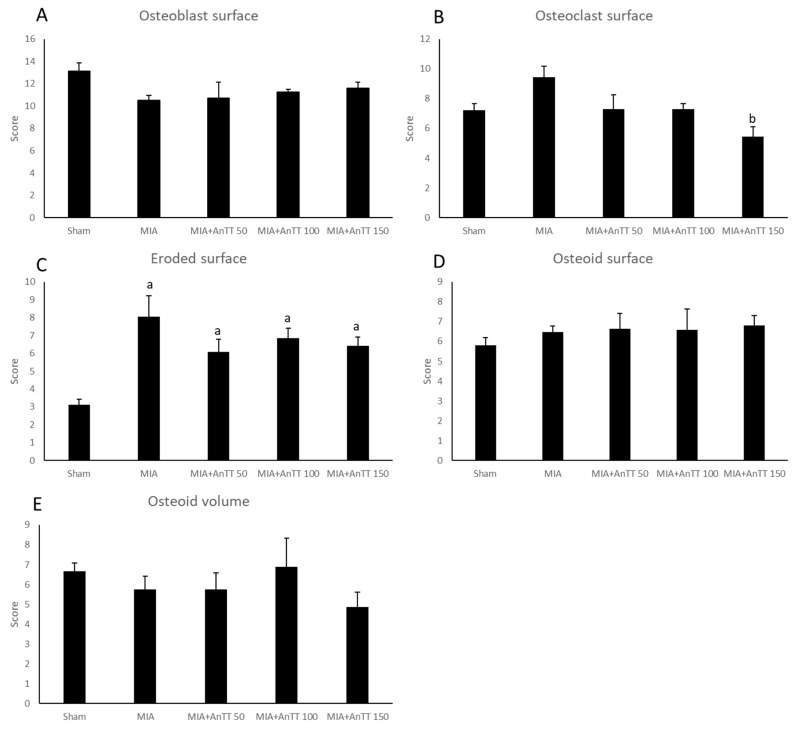
Cellular indices of subchondral bone after treatment: osteoblast surface (**A**); osteoclast surface (**B**); eroded surface (**C**), osteoid surface (**D**) and osteoid volume (**E**). The data are shown as mean with standard error of the mean. Letter ‘a’ indicates significant difference versus the sham group; ‘b’ versus the MIA group. Abbreviation: AnTT, annatto tocotrienol.

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
