# Peer review of "The Effects of Annatto Tocotrienol Supplementation on Cartilage and Subchondral Bone in an Animal Model of Osteoarthritis Induced by Monosodium Iodoacetate"

_ijerph, 2019, doi:10.3390/ijerph16162897_

Round 1

Reviewer 1 Report

The authors answered my comments and improved the manuscript with revision.

Author Response

Thank you for your positive reply. There is no comment to be addressed. 

Reviewer 2 Report

Authors improved the quality of the manuscript by answering to comments. The article can be accepted in its current form.

Author Response

Thank you for your positive reply. We appreciated your comments in improving this manuscript. 

Reviewer 3 Report

This is a revised manuscript testing the effect of annatto tocotrienol (AT) on OA progression in rats. Based on the findings presented the authors conclude that AT protects cartilage against degradation using monosodium iodoacetate OA model. While these findings could be of potential interest, the quality of presentation and the figures shown are not very convincing and do not support the conclusion.

Specific Points:

The micrographs shown in Figure 2 are of bad quality and it is not clear whether these sections are from a similar depth in the tissue. In addition, it is not clear what grading system the authors have used. More importantly the synovium is not shown in the histological figures and it is not clear how the authors identified inflammatory cells.Furthermore, it is not clear why the degree of cartilage degradation was not scored according to the OARSI scoring system.

Figure 4: The use of HA in serum to use as a biomarker for OA is rather strange and can not be done since HA is been g released by many tissues in the body.

Author Response

Thank you for the comments. Please kindly refer to the attached response sheet for our reply. The changes in the text have been highlighted in yellow. 

Thank you. 

Reviewer 4 Report

The authors provide little evidence to support their overall conclusion that “Annatto tocotrienol exerts joint protective effects by preventing cartilage degradation and subchondral remodelling in an animal model of osteoarthritis. The optimal dose of annatto tocotrienol is suggested to be 100 mg/kg/day, as it showed the most favourable changes in the joint histology and serum cartilage markers.” The histological analysis in figure 2 suggests that AnTT has a protective role in MIA, but it is difficult to interpret the results due to the poor quality of the images. Figure 3 shows that AnTT decreases circulating HA levels compared to MIA, however AnTT increases bone erosion markers (specifically CTX). While COMP levels are affected by the various treatments, COMP is not a marker of cartilage breakdown, but of cartilage turnover, which can include de novo synthesis. Figure 5 shows that AnTT has no protective effect in MIA treated animals which the exception of one dose for one parameter. Particularly concerning is that the erosion surface remains unchanged by AnTT treatment, which contradicts figure 2. Furthermore, the authors failed to adequately address many of my major concerns about their manuscript. In summary, the current data does not provide enough evidence to reach the conclusion that AnTT has a protective effect on cartilafge degeneration and bone erosion in the MIA model.

Reviewer Comment: Second, the article contains no histological images, despite that Figure 1 and 3 are entirely based on histological analyses. This makes it impossible for the reader to make their own judgement with regard to histological changes that are reportedly occurring.

Author Reply: Thank you for the comment. We have added the micrographs of the joint as Figure 2 (page 5).

Reviewer reply: The authors have added histological images corresponding to the analysis in Figure 3. However, the orientation of each image is different and in some cases show only partially overlapping areas between images. For example the sham and MIA images are rotated differently and show unequal amounts of each side of the joint between images. Images should be orientated in the same direction, show the same proportion of the tibia and femur, and the tibia and femur should be indicated. Scale bars should also be defined in the figure legend. Images corresponding to the analysis in figure 5 are still missing and needed to be added. The methods paper the authors cited used a much higher magnification for their analysis, presumably the authors of the current study did the same and thus need to provide representative images.

Reviewer Comment: Third, based on figure 2, the authors’ conclusions seem inherently flawed. The normal control group, which received injections of saline and olive oil, but not MIA, both have increased levels of serum HA and CTX. This suggests there was either something wrong with the experimental setup or that these assays are prone to large variation and error. Regardless, it makes any potential drug effects difficult to interpret.

Author Reply: Thank you for the concern. We acknowledge the increase in circulating HA and CTX level in the rats at the end of the treatment period, but this alternation also happened to the AnTT-treated groups. Since the samples were processed in a single batch to minimize the errors, we suggest that the increase could be caused by age. Since the rats used in the current study was relatively young, the increase of these markers might reflect an active modelling of the musculoskeletal system.

Reviewer reply: This explanation does not alleviate my concerns and in fact only raises new problems with the paper.

If the authors think these changes are age-dependent, then why did they not use skeletally mature mice?

Moreover, this explanation does not address the fact that CTX is elevated in all of the MIA+AnTT-treated groups compared to MIA after 5 weeks of treatment, which suggests the AnTT increases bone resorption compared to MIA alone. The sham group also appears to have increased serum CTX compared to MIA, although not indicated. This is quite surprising considering that the Sham group mean value of CTX is higher and the variation is lower compared to MIA+AnTT50, which is significantly different from MIA. The authors need to explain this and should include mean and variation throughout the text for key results.

Reviewer Comment: Regardless, it makes any potential drug effects difficult to interpret. Furthermore, annatto tocotrienol appears to increase bone erosion based on the increased serum level of CTX in all treatment groups compared to negative control MIA rats. These findings are at odds with the authors’ conclusions that tocotrienol may have a positive effect for OA

Author Reply: Thank you for the concern. We have noted the effects of AnTT on bone CTX markers and altered the conclusion to remind the readers to be cautious of this finding. “However, the effects of annatto tocotrienol on subchondral bone remodelling should be interpreted with cautions as it does not prevent the increase in serum bone resorption marker but lowers bone formation marker at high dose”

(line 295–297)

Reviewer reply:

Reviewer Comment: In figure 1 what do the x’s and circles represent.

Author Reply: Thank you for the reminder. We have

added the following at the legend of Figure 3 (renamed from Figure 1):

 “The symbol ‘x’ in the box plot represents the mean value, ‘’ represents the outliers, while the vertical line in the box represents the median of the group.” (line 172–173)

Reviewer reply: How is the variation in figure 3 shown? How were the outliers calculate? There appear to be outliers that fall within the SEM, which is then unlikely to be an outlier? The authors state in the figure legend that the median is indicated by the vertical line but perhaps they mean the horizontal line?

Reviewer Comment: Figure 3 is missing letter labels.

Author Reply: The letter labels are indicated at the footnote of the Figure 5 (renamed from Figure 3) (line 200–201).

Reviewer reply: The labels for the graphs are still missing.

Reviewer Comment: It is unclear how the parameters in figure three were evaluated form the methods section.

Author Reply: Thank you for the comments. We have elaborated on the measurements of bone cellular histomorphometry results depicted in Figure 5 (renamed from Figure 3)

(line 115–122).

Reviewer reply: The authors need to provide example images illustrating how they determined each kind of surface. Particularly concerning is how an ‘osteoclast surface’ is defined since osteoclasts are very thin cells and are difficult to identify without specific histological stains like TRAP.

Reviewer Comment: The authors used only male rats.

Author Reply: Thank you for the comment. Male rats were used in this study to prevent the influence of fluctuation in oestrogen level on cartilage and bone metabolism. However, we acknowledge that osteoarthritis is more common in elderly women. This has been noted as the limitation of the study

(line 290–292).

Reviewer reply: This is really not a valid reason to only use male rats, especially when osteoarthritis is common in women. There is little evidence to suggest that natural flucation of estrogen levels will impact the results and many studies on bone erosion use female or male and female rats. While it is unreasonable for the authors to have to repeat all of the experiments in female rats at this point, they need to at least have a proper discussion of this major limitation.

Author Response

(The authors gave the same response as above.)

Round 2

Reviewer 3 Report

Unfortunately the authors did not adequately address the concerns I raised in my previous review. Most importantly, the histological figures are still of bad quality and clearly the sections shown are not in the same depth. For example, articular cartilage in the section of the sham knee joint looks similar to the articular cartilage in the MIA-treated knee joint. Furthermore, subchondral bone has to be quantitatively analyzed by microCT or by bone histomorphometry software. Finally, the quantification of immune cells should be done by immunohistochemistry. These issues are major concerns and cannot just be addressed with limitations of this study.

Author Response

Thank you for your comment. Please kindly find our replies in the attached response sheet. 

We look forward to your positive response. 

Reviewer 4 Report

The authors have addresses most of my concerns. It is now easier to follow the results and the discussion has been revised to address the fact that the results provide weak support for Annatto Tocotrienol  in the MIA model. 

Author Response

Thank you for the constructive comments. No replies are required from us. 

This manuscript is a resubmission of an earlier submission. The following is a list of the peer review reports and author responses from that submission.

Round 1

Reviewer 1 Report

In the present manuscript the authors study the anti-osteoarthritis effects of a natural molecule, annatto tocotrienol supplementation on cartilage and subchondral bone. They use a MIA-induced osteoarthritis model, which is a pertinent in vivo model in rats. By convincing in vivo experiments (histological and biochemical evaluations), they demonstrate that oral annatto tocotrienol supplementation may retard the progression of osteoarthritis.
In addition, it is a well-written article and covers an interesting topic.

Minor comment: images from histological analyses may be added as supplemented figures.

Reviewer 2 Report

This manuscript is addressed to evaluate the effectiveness of oral supplementation of Annatto Tocotrienol on the tibiofemoral joint in a rat model of osteoarthritis (OA), a well known model of OA-like pain. It is an interesting study as it evaluates through histological, and serum assessments what changes occur after the oral supplementation of Annatto Tocotrienol, well known to have anti-inflammatory properties. The article is well written, and references are proper.

Major revisions:

I would suggest to Authors to specify in the title what types of effects occur following oral supplementation of Annatto Tocotrienol because it is too generic and also the specific animal model used for this study.

How did the Authors select the concentration of Annatto Tocotrienol for the oral supplementation? Did the Authors know what the bioavalaibility of this compound is within the articular joint? Did the Authors think of delivering this compound intra-articularly in the knee joint to ensure a direct action on the injured articular areas?

Page 2 line 82. Please report the mean mass of rats

Page 3 lines 97-112. Please add further details on this section as suggested below.

Please report the time of formalin fixation.

Please specify the cutting plane level (longitudinal ?) used for histological assessment.

Please provide a reference for haematoxylin/Eosin and Safranin stainings, otherwise, provide a brief explanation with these methods.

Why did the authors carry out decalcification for two months? Such a long time of decalcification could alter bone microarchitecture thus creating technical biases for the assessment of the bone component. What do the Authors think?

Apart from histological assessment, did the authors perform immunohistochemical analyses on catabolic and apoptotic markers to assess the effectiveness of Annatto Tocotrienol?

Page 3 line 129. Apart from bar graphs, the Authors should insert histological micrographs performed on the tibiofemoral joint to appreciate characteristic histological features for each experimental/control group.

Page 7 line 177. Authors should report the limitations of this study in this section.

Reviewer 3 Report

This manuscript investigated how Annatto Tocotrienol (AT) supplementation in the drinking water affects OA progression in rats. The authors conclude that AT supplementation slows down cartilage degradation and subchondral bone changes in rats in which OA was induced by MIA. Since no histology figures are shown in this manuscript, one cannot judge whether indeed AT slowed down cartilage degradation and inhibited subchondral bone changes. It is somewhat surprising that the error bars especially in Figures 2 and 3 are relatively small especially considering the nature of the experiments. Especially, the conclusion that AT affects subchondral bone changes is not supported by the findings.

Special Points:

The terms positive and negative controls should not be used in this context.

Reviewer 4 Report

The authors investigated the effects of Annatto tocotrienol on bone-related parameters in the MIA model of OA. They conclude that tocotrienol may reduce OA progression and reduce bone erosion caused by MIA injection. However, there are several issues that make the manuscript unsuitable for publication. First, I am not sure the article falls within the scope of the journal. It may be more suited in another MDPI journal such as IJMS. Second, the article contains no histological images, despite that Figure 1 and 3 are entirely based on histological analyses. This makes it impossible for the reader to make their own judgement with regard to histological changes that are reportedly occurring. Third, based on figure 2, the authors’ conclusions seem inherently flawed.  The normal control group, which received injections of saline and olive oil, but not MIA, both have increased levels of serum HA and CTX. This suggests there was either something wrong with the experimental setup or that these assays are prone to large variation and error. Regardless, it makes any potential drug effects difficult to interpret. Furthermore, annatto tocotrienol appears to increase bone erosion based on the increased serum level of CTX in all treatment groups compared to negative control MIA rats. These findings are at odds with the authors’ conclusions that tocotrienol may have a positive effect for OA. 

Minor issues

1.    More information about tocotienol preparation is required. 

2.    What concentration of EDTA was used?

3.    In figure 1 what do the x’s and circles represent. 

4.    Figure 3 is missing letter labels

5.    It is unclear how the parameters in figure three were evaluated form the methods section. 

6.    The authors used only male rats